# Robust Fact-Checking under Contaminated Evidence Sources via Claim Decomposition and Dynamic Reweighting

## Abstract

Fact-checking seeks to assess the truth values of claims with respect to a knowledge base from which supporting or refuting evidence can be retrieved. However, most existing approaches assume access to a clean and reliable knowledge source. In practice, retrieved evidence is often contaminated with misinformation, which substantially reduces verification accuracy. In this paper, we address the task of fact checking under contaminated knowledge bases and propose a framework designed to remain robust in noisy environments. Our approach first decomposes each claim into subclaims, then retrieves candidate evidence for each subclaim. A large language model (LLM) is subsequently employed to classify documents into supporting, refuting, or unrelated categories, and subclaim veracity is determined through a carefully weighted majority stance. To further enhance robustness, documents are dynamically reweighted: supporting evidence is upweighted as likely truthful, while refuting evidence is downweighted as potentially misleading, and these weights are incorporated into subsequent retrieval through reranking. To rigorously evaluate this setting, we also introduce a method for constructing adversarially contaminated knowledge bases by generating misinformation derived from gold evidence and false claims, which effectively misleads standard retrievers. Experimental results across open-source LLMs and datasets demonstrate that contamination severely degrades baseline fact checking performance, while our framework substantially mitigates this effect.

## 1 Introduction

The proliferation of misinformation presents substantial challenges to reliable information access. Automated fact-checking has emerged as a critical tool for identifying false claims, with recent systems frequently integrating retrieval mechanisms (Lewis et al., 2020; Thorne et al., 2018) and large language models (LLMs) to evaluate claim veracity. Nevertheless, a fundamental limitation persists across much of this research: the assumption that the underlying knowledge base (KB) is clean and trustworthy. In real-world settings, however, KBs are often contaminated, containing a mixture of accurate, misleading, and fabricated documents. When such misinformation is retrieved as evidence, fact checking accuracy deteriorates significantly, underscoring the urgent need for robustness in fact-checking systems.

In this work, we formalize the problem of fact checking under contaminated knowledge bases and propose a framework to address it. The method begins by decomposing complex claims into simpler subclaims, followed by the retrieval of relevant documents for each subclaim. A large language model (LLM) then categorizes the retrieved documents according to stancesupporting, refuting, or unrelated. Subclaim veracity is determined by a weighted majority, wherein supporting evidence outweighs refuting evidence. This classification further informs a reliability update: supporting documents are upweighted as likely truthful evidence, while refuting documents are downweighted as potential misinformation. These dynamic weights are incorporated into subsequent retrieval through candidate reranking, iteratively steering the system toward trustworthy sources while suppressing misleading ones.

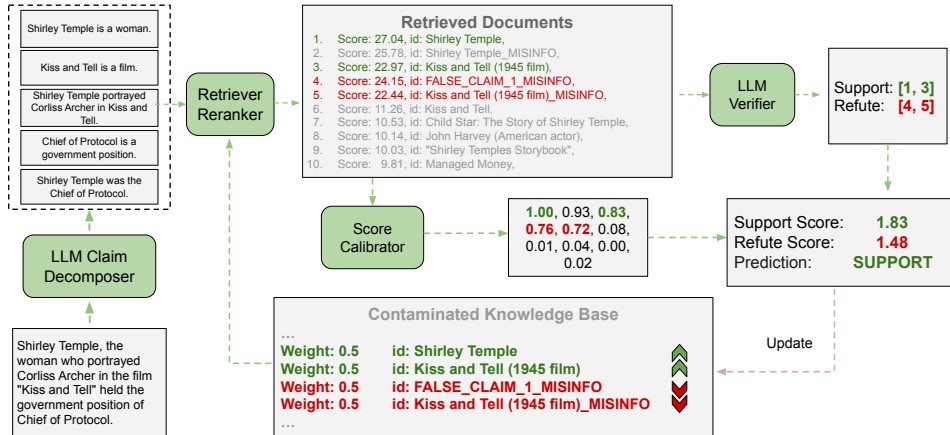

Figure 1: An overview of our method. A complicated claim is first decomposed into multiple simple subclaims. For each subclaim, a retriever will retrieve top k (here k=10) most related documents to that claim. Then, an LLM Verifier will do reasoning, indicating the stance of each document toward the subclaim.

To ensure a realistic and challenging evaluation, we develop a method for constructing knowledge bases contaminated with misinformation. In particular, misinformation is derived from both true and false claims, yielding adversarial distractors that are fluent and topically relevant. This contamination procedure consistently degrades the performance of baseline retriever-augmented fact checking frameworks, demonstrating its effectiveness in stressing current systems.

We evaluate the proposed framework using four open-source LLMs including Qwen3-14B (Yang et al., 2025), Llama-3.1-8B-Instruct (Dubey et al., 2024), Gemma-3-12b-it (Team et al., 2025), and Mistral-7B-Instruct-v0.3 (Chaplot, 2023), across six datasets: HoVer (Jiang et al., 2020), EX-FEVER (Ma et al., 2023), HotpotQA (Yang et al., 2018), SciFact (Wadden et al., 2020), Pub-Health (Kotonya & Toni, 2020), and Climate-FEVER (Diggelmann et al., 2020). Experimental results demonstrate that contamination markedly reduces the accuracy of baseline fact checking methods, whereas our approach substantially recovers performance by amplifying reliable evidence and mitigating the influence of misinformation.

Our contributions are threefold:

1. **Misinformation-contaminated knowledge base construction.** We propose an effective method for generating contaminated datasets by deriving misinformation from gold evidence and false claims. This procedure yields adversarially strong distractors that substantially degrade the performance of existing retrieval-based fact checking systems, thereby enabling rigorous evaluation under realistic noisy conditions. The resulting misinformation-augmented datasets, built upon multiple fact-checking and QA benchmarks, will be released publicly.
2. **LLM-based stance aggregation with iterative weight updating.** We introduce a framework that employs large language models to classify retrieved documents according to stance toward a subclaim (supporting, refuting, or unrelated) and to determine subclaim veracity via majority aggregation. The stance signals are further used to iteratively adjust document reliability weights, which are incorporated into retrieval reranking to suppress misinformation while amplifying trustworthy evidence.
3. **Comprehensive empirical evaluation.** We conduct extensive experiments with four open-source LLMs across six benchmark datasets. Results show that contamination induces substantial performance drops in standard baselines, while our framework consistently mitigates this degradation, demonstrating robustness and generalizability across diverse domains.

## 2 RELATED WORKS

Research on reliable fact-checking systems spans several interconnected areas, including fact checking, misinformation detection, and robust retrieval from noisy knowledge sources. Fact checking examines how to assess the veracity of a claim given external evidence, while misinformation detection focuses on identifying misleading or adversarial content that may distort reasoning. These two directions are inherently complementary: fact checking depends on accurate evidence selection and logical inference, whereas misinformation detection enhances the robustness of this process when

the evidence pool is corrupted. Our work lies at the intersection of these areas, explicitly addressing fact checking under contaminated knowledge bases, where evidence retrieval is complicated by the presence of adversarial misinformation.

## 2.1 FACT CHECKING

Automated fact-checking has been a longstanding focus within natural language processing. Benchmarks such as FEVER, HoVer, and EX-FEVER provide large-scale collections of labeled claims with associated gold evidence, facilitating systematic evaluation of both retrieval and reasoning components. Early approaches primarily relied on supervised classifiers to predict claim veracity based on retrieved sentences, whereas recent work increasingly leverages retrieval-augmented large language models (LLMs), which are capable of jointly reasoning over claims and evidence.

Complex claims that require multi-hop reasoning have motivated the development of claim decomposition methods. For instance, GraphCheck (Jeon & Lee, 2025) represents claims as entity-relation graphs and verifies them by decomposing them into subclaims, which are then evaluated in a structured manner. This approach improves both accuracy and interpretability on multi-hop benchmarks. Similarly, ProgramFC (Pan et al., 2023) decomposes claims into executable programs for fact-checking. However, the performance of these methods degrades sharply when the underlying knowledge base contains misinformation. Our framework also incorporates claim decomposition as a preprocessing step but shifts the focus to robustness: how to verify claims effectively when the knowledge base itself is contaminated by misinformation—a challenge that has been largely overlooked in prior research.

## 2.2 MISINFORMATION DETECTION

Misinformation detection focuses on identifying false or misleading content within corpora or retrieved contexts. A widely adopted approach is stance classification, in which documents are categorized as supporting, refuting, or unrelated to a claim, and the aggregated stance distribution serves as a proxy for veracity assessment. Many fact-checking pipelines integrate stance detection as an intermediate component, enabling systems to strengthen conclusions through consistent evidence while attenuating the influence of conflicting signals.

Another line of research explores robustness under adversarial contamination. Weller et al. (2022) demonstrate that injecting poisoned documents into open-domain QA corpora substantially degrades model performance, and propose redundancy-based methods to improve robustness by detecting consistent answers across contexts. Their approach employs query rewriting to increase the density of retrieved evidence but does not incorporate feedback mechanisms after decision-making. Moreover, their method constructs misinformation solely by applying SpaCy NER (Honnibal, 2017) to identify entities and replacing correct answers, resulting in limited diversity of adversarial examples.

Our work bridges these research directions by explicitly addressing fact-checking under contaminated knowledge bases. Unlike prior fact-checking systems, we consider scenarios in which misinformation is deliberately introduced into the evidence pool. In contrast to earlier contamination-robust QA approaches, our framework leverages LLM-based stance classification and iterative reweighting to dynamically adjust document influence, thereby suppressing misinformation while amplifying reliable evidence during fact checking. Furthermore, our knowledge base poisoning strategy generates diverse and adversarially strong misinformation capable of misleading retrievers, yielding a more rigorous and challenging evaluation setting.

## 3 METHODS

We introduce the novel task of fact-checking in the presence of a contaminated knowledge base and propose a robust method for addressing this adversarial setting. Section 3.1 formally defines the problem setting. Section 3.2 provides an overview of the claim decomposition strategy employed and discusses its potential for enhancing robustness against misinformation. Section 3.3 details our approach for constructing effective and topically consistent misinformation, designed to impose greater challenges for fact-checking systems. Finally, Section 3.4 presents our stance-grouping and

evidence reweighting strategy, which enables more reliable verification by dynamically refining the evaluation of retrieved evidence.

### 3.1 PROBLEM FORMULATION

Given a set of claims $C$ and a contaminated knowledge base $\tilde{K}$, our framework integrates a large language model $V$ as a verifier, and a search engine $R$ as retriever. This framework aims to predict a label $Y$ to evaluate the claim as TRUE or FALSE, based on $\tilde{K}$. We base our approach on two key observations: 1) Misinformation typically constitutes only a small portion of a knowledge base. 2) Large corpora, such as Wikipedia, generally contain substantial redundancy in their information; in other words, supporting evidence to a claim might be found in multiple documents.

### 3.2 CLAIM DECOMPOSITION

Prior work such as GraphCheck (Jeon & Lee, 2025) has shown the effectiveness of decomposing claims into subcomponents for multi-hop reasoning. Inspired by this idea, our framework employs a large language model (LLM) to decompose each original statement into simpler subclaims. The decomposition reduces reasoning complexity and broadens the scope of related documents. In dense corpora such as Wikipedia, true claims are typically supported by a larger volume of consistent evidence than false ones, and retrieving redundant but related documents substantially aids in determining veracity. Thus, claim decomposition serves as a

---

**Algorithm 1** Access and Update Weight

**Input:** weight map $W$, evidence $e$, prediction $d$, existence $\sigma$, retrieval score $\rho$

    **function** UPDATEWEIGHT$(e, d, \sigma, \rho)$
        **if** $d = \sigma$
            $W[e].pos \leftarrow W[e].pos + \rho$
        **else**
            $W[e].neg \leftarrow W[e].neg + \rho$

    **function** GETWEIGHT$(W, e)$
        $p, n \leftarrow W[e].pos, W[e].neg$
        **return** $\dfrac{1}{1 + e^{n-p}}$

---

critical preprocessing step in our method, enabling more robust verification under noisy knowledge bases. We provide the prompt for claim decomposition in Appendix B.4.

### 3.3 MISINFORMATION KNOWLEDGE BASE CONSTRUCTION

Designing effective misinformation is a non-trivial challenge. Naive strategies, such as simply negating the original evidence, often yield examples that lack diversity and contextual relevance, making them easily filtered out by retrievers and thus ineffective for rigorously testing fact checking systems. Similarly, entity replacement approaches, such as the method proposed by Weller et al. (Weller et al., 2022), use SpaCy NER (Honnibal, 2017) to substitute the answer in a QA datapoint with another named entity. While this technique can generate superficially modified claims, it often produces statements that are misleading rather than strictly false. This limitation is especially pronounced for open-ended claims. For instance, replacing *painter* with *engineer* in the claim *"Leonardo da Vinci is a painter"* does not result in a false statement, as da Vinci was both a painter and an engineer. Our experiments further confirm that

---

**Algorithm 2** Subclaim Verification

**Input:** contaminated knowledge base $\tilde{K}$, retriever Retrieve, verifier Verify, weight map $W$, subclaim $c$

    **function** VERIFYSUB-
CLAIM$(c, \tilde{K}, \text{Retrieve}, \text{Verify}, W)$
        $(E, \rho) \leftarrow \text{Retrieve}(c, \tilde{K}, W)$
        $(S, F, U) \leftarrow \text{Verify}(E, c)$
        $s^{+} \leftarrow \sum_{e \in S} \rho(e) \cdot \text{GETWEIGHT}(W, e)$
        $s^{-} \leftarrow \sum_{e \in F} \rho(e) \cdot \text{GETWEIGHT}(W, e)$
        $d \leftarrow \begin{cases} +1, & \text{if } s^{+} \geq s^{-} \\ -1, & \text{otherwise} \end{cases}$
        **for all** $e \in S \cup F$
            $\sigma(e) \leftarrow \begin{cases} +1, & \text{if } e \in S \text{ (supporting)} \\ -1, & \text{if } e \in F \text{ (refuting)} \end{cases}$
            UPDATEWEIGHT$(e, d, \sigma(e), \rho(e))$
        **return** $d$

---

such entity-replacement strategies do not substantially degrade the performance of fact checking systems when operating over contaminated knowledge bases.

To overcome these limitations, we leverage the structured annotations available in many fact checking and QA datasets. Such resources often include gold-standard evidence, entity identifiers, and

explicit truth labels, which provide opportunities to construct misinformation that is both semantically consistent and more challenging for retrieval and verification models. In addition, fact checking datasets contain numerous human-authored **False** claims, which naturally serve as a rich source of misinformation. By paraphrasing these claims or generating "supporting evidence" for them, we can construct more diverse, contextually grounded, and adversarially strong misinformation.

We propose two complementary strategies for misinformation generation, tailored to true and false claims, respectively. A central design principle in both strategies is robustness against document retrievers. To ensure this, we preserve the gold entities and evidence so that the generated misinformation remains topically aligned with the original statements. 1) For claims labeled as **true**, we prompt a large language model (LLM) with the original claim, its gold evidence, and the relevant entities. The LLM is instructed to reuse these entities in new but natural contexts, thereby producing misinformation that is coherent, entity-consistent, and substantially more challenging than simple negations or substitutions. 2) For claims labeled as **false**, we instead instruct the LLM to generate fabricated supporting evidence that incorporates the gold entities. This results in plausible yet misleading justifications that are topically consistent with the original claim, thereby increasing the difficulty of verification.

Our approach produces misinformation that is both diverse and retriever-resilient. On average, the misinformation ratio per retrieval reaches 20 - 30%, approximately two to three times the volume of the original gold evidence. Moreover, in more than 90% of cases where gold evidence is retrieved, its misinformation counterpart is also retrieved. This high rate of co-occurrence indicates that the generated misinformation is effectively integrated into the retrieval process, creating a more rigorous and adversarial testbed for fact checking models.

### 3.4 FACT CHECKING WITH DOCUMENT WEIGHT UPDATING

We begin by assigning a default weight to each document. For a given claim, we first decompose it into a set of simpler subclaims. For each subclaim, we retrieve a collection of related documents along with their retrieval scores. These documents are concatenated to form a candidate evidence text, which is then processed by a large language model (LLM) to classify

---

**Algorithm 3** Full Claim Verification

**Input:** contaminated knowledge base $\tilde{K}$,
retriever Retrieve, verifier Verify, weight map $W$, claim $c$

    $subclaims \leftarrow \text{DECOMPOSE}(c)$
    **for** $c_s$ in $subclaims$
        **if not** $\text{VERIFYCLAIM}(c_s, \tilde{K}, \text{Retrieve}, \text{Verify}, W)$
            **return** False
    **return** True

---

each document according to its stance toward the subclaim: support, refute, or unrelated. Then, we compute a final score for the subclaim by integrating document weights, retrieval scores, and stance labels. If the subclaim is judged to be true, the weights of supporting documents are increased proportionally to their retrieval scores and current weights, while the weights of refuting documents are decreased. Conversely, if the subclaim is judged to be false, refuting documents are upweighted and supporting documents are downweighted (see Algorithm 1, 2). Finally, the original claim is classified as True if and only if all of its subclaims are identified as True (see Algorithm 3).

In dense knowledge bases such as Wikipedia, we assume that a subclaim is likely to be true if the number of supporting evidences exceeds the number of refuting evidences. Once a subclaim is identified as True, supporting evidences are treated as reliable, while refuting evidences are treated as unreliable. We then update the weight of each document by integrating its stance assignment with its retrieval score. These updated weights are normalized by a sigmoid function and then used to re-rank documents in subsequent retrieval iterations, thereby promoting sources deemed more trustworthy and demoting those assessed as less credible.

Each document weight entry has two fields $pos$, and $neg$ both initialized as 0, which are iteratively updated during verification. When these weights are utilized, they are passed through a sigmoid function (Algorithm 1) to ensure that the resulting confidence values remain within the interval [0, 1]. As a secondary outcome, the iterative document weight updating process offers a principled mechanism for inferring the reliability of individual documents over the course of fact checking.

| Datasets | Mode | Qwen3 14B | Gemma-3 12b-it | Llama-3.1 8B-Instruct | Mistral-7B Instruct-v0.3 |
|---|---|---|---|---|---|
| HoVer | Clean | 66.15 | 63.58 | 66.44 | 61.03 |
| | Misinfo Baseline | 50.15 | 48.86 | 42.34 | 49.86 |
| | Misinfo Weight Update | 58.33 (+8.18) | 57.10 (+8.24) | 55.43 (+13.09) | 56.05 (+6.19) |
| EX-FEVER | Clean | 68.43 | 62.20 | 63.06 | 64.29 |
| | Misinfo Baseline | 46.28 | 47.39 | 44.74 | 49.69 |
| | Misinfo Weight Update | 58.71 (+12.43) | 60.48 (+13.09) | 58.63 (+13.89) | 58.26 (+8.57) |
| HotpotQA | Clean | 72.18 | 71.07 | 67.05 | 67.69 |
| | Misinfo Baseline | 48.71 | 48.49 | 49.03 | 43.66 |
| | Misinfo Weight Update | 59.93 (+11.22) | 56.43 (+7.94) | 60.12 (+11.09) | 54.64 (+10.98) |
| PUBHEALTH | Clean | 64.23 | 54.86 | 60.91 | 56.99 |
| | Misinfo Baseline | 52.76 | 48.66 | 52.68 | 51.39 |
| | Misinfo Weight Update | 60.06 (+7.30) | 52.63 (+3.97) | 58.52 (+5.84) | 55.06 (+3.67) |
| SciFact | Clean | 63.89 | 62.38 | 69.48 | 73.41 |
| | Misinfo Baseline | 49.11 | 44.13 | 43.37 | 49.32 |
| | Misinfo Weight Update | 61.78 (+12.67) | 60.80 (+16.67) | 63.76 (+20.39) | 64.36 (+15.04) |
| Climate-FEVER | Clean | 66.26 | 62.54 | 64.75 | 65.93 |
| | Misinfo Baseline | 34.82 | 28.98 | 31.29 | 37.93 |
| | Misinfo Weight Update | 49.28 (+14.46) | 46.01 (+17.03) | 56.78 (+25.49) | 51.27 (+13.34) |

Table 1: Macro-F1 scores under open-book settings for each dataset benchmark and backbone LLM. Clean means we conduct experiments on clean knowledge base, as a ideal upper bound. Misinfo Baseline means we run baseline method on contaminated knowledge bases. Misinfo Weight Update means we run our method on contaminated knowledge bases.

| Datasets/Models | GraphCheck Clean | GraphCheck Misinfo | ProgramFC Clean | ProgramFC Misinfo |
|---|---|---|---|---|
| HoVer | 65.15 | 51.21 | 61.60 | 46.63 |
| EX-FEVER | 64.92 | 45.78 | 66.40 | 42.82 |
| HotpotQA | 70.14 | 49.71 | 72.48 | 27.28 |
| PUBHEALTH | 63.77 | 50.73 | 63.48 | 50.87 |
| SciFact | 65.49 | 49.86 | 70.69 | 42.97 |
| Climate-FEVER | 61.06 | 37.74 | 63.06 | 32.94 |

Table 2: Macro-F1 scores of other frameworks. The performance dropped drastically under a misinformation contaminated knowledge base.

## 4 EXPERIMENTS

### 4.1 EXPERIMENTAL SETUP

**Datasets and Benchmarks.** We conduct experiments on six benchmark datasets: HoVer, EX-FEVER, HotpotQA, PubHealth, SciFact, and Climate-FEVER. Among them, HoVer and EX-FEVER are fact checking datasets in the open domain, while PubHealth, SciFact and Climate-FEVER focus specifically on public health, science and climate, respectively. All these datasets contain human-written false claims, which are already diverse and rich misinformation and can be helpful in our misinformation construction. HotpotQA is originally a multi-hop question answering dataset in the open domain. To adapt it for fact checking, we preprocess each instance by combining the question and its answer into a declarative form, and then prompt a large language model (LLM) to generate both true and false statements with respect to the gold evidence.

For each clean dataset, we build a contaminated counterpart by injecting two types of misinformation documents: 1) A refutation that contradicts to every gold evidence labeled in the original dataset. 2) Fake evidence that provide *supportive* evidence to every false claim. In such setting, we provide misinformation more than gold evidence to every claim in its original dataset, making the task more challenging. We also provide statistics for retrieval frequency in misinformation settings.

| Datasets | Retrieval | Qwen3 14B | Gemma-3 12b-it | Llama-3.1 8B-Instruct | Mistral-7B Instruct-v0.3 |
|---|---|---|---|---|---|
| HoVer | Total | 338770 | 348750 | 346790 | 327560 |
| | Misinfo | 89012 (26.28%) | 90104 (25.84%) | 88525 (25.53%) | 100439 (30.66%) |
| | Golden | 33795 (9.98%) | 31649 (9.07%) | 30841 (8.89%) | 32289 (9.86%) |
| EX-FEVER | Total | 368010 | 376820 | 375350 | 365810 |
| | Misinfo | 53671 (14.58%) | 52053 (13.81%) | 51253 (13.65%) | 57895 (15.83%) |
| | Golden | 27843 (7.57%) | 24986 (6.63%) | 25100 (6.69%) | 26999 (7.38%) |
| HotpotQA | Total | 442380 | 450150 | 423030 | 416700 |
| | Misinfo | 67723 (15.31%) | 69144 (15.36%) | 68766 (16.26%) | 76019 (18.24%) |
| | Golden | 37967 (8.58%) | 36096 (8.02%) | 36124 (8.54%) | 37493 (9.00%) |
| PUBHEALTH | Total | 24170 | 25110 | 26760 | 21540 |
| | Misinfo | 6618 (27.38%) | 7251 (28.88%) | 7303 (27.29%) | 6754 (31.36%) |
| | Golden | 2107 (8.72%) | 2197 (8.75%) | 2301 (8.60%) | 1998 (9.28%) |
| Climate-FEVER | Total | 84200 | 82680 | 76470 | 79590 |
| | Misinfo | 30214 (35.88%) | 28765 (34.79%) | 26328 (34.43%) | 30250 (38.01%) |
| | Golden | 3800 (4.51%) | 3540 (4.28%) | 3124 (4.09%) | 3637 (4.57%) |

Table 3: Retrieval statistics. In our misinformation setting, number of misinfo retrievals are much more than number of gold retrievals. However, thanks to redundancy of the knowledge base, those non-gold but true documents still provide some support towards the truth.

| Misinfo Datasets | Qwen3 14B | Gemma-3 12b-it | Llama-3.1 8B-Instruct | Mistral-7B Instruct-v0.3 |
|---|---|---|---|---|
| HoVer | 57.86 | 59.30 | 49.59 | 58.53 |
| EX-FEVER | 44.99 | 55.25 | 47.24 | 54.73 |
| HotpotQA | 54.97 | 59.21 | 51.36 | 54.94 |
| PUBHEALTH | 61.60 | 34.10 | 50.45 | 67.29 |
| SciFact | 51.27 | 46.64 | 48.79 | 49.31 |
| Climate-FEVER | 66.88 | 68.39 | 63.33 | 65.99 |

Table 4: Macro F1 Scores of classification on misinformation documents.

The datasets also differ in their underlying knowledge sources. HoVer, EX-FEVER, and HotpotQA rely on a Wikipedia dump, which provides a dense and information-rich knowledge base. In contrast, PubHealth, SciFact and Climate-FEVER are grounded in curated knowledge bases, tailored to their respective domains.

**Backbone Models.** We evaluate our approach using four state-of-the-art open-source LLMs as backbone models for the verifier: Qwen3-14B, Llama-3.1-8B-Instruct, Mistral-7B-Instruct-v0.3, and Gemma-3-12B-it. For the original claim decomposition task, we use Qwen2.5-72B-Instruct as recommended in GraphCheck.

**Experimental Design.** For each (dataset, backbone) combination, we perform three experimental settings: 1) Baseline on **clean** knowledge base: evaluating model performance without contamination. 2) Baseline on **contaminated** knowledge base: evaluating the effect of injected misinformation. 3) Our method on **contaminated** knowledge base: applying stance grouping and weight updating to mitigate contamination.

We index each corpus twice: 1) a lexical index (BM25) (Robertson et al., 2009), tuned with $k_1 = 0.9, b = 0.4$; and 2) a dense FAISS index (Douze et al., 2024) with the intfloat/e5-base-v2 embedding model (Wang et al., 2022). We then use the hybrid retriever with $\alpha = 0.5$ to retrieve top 10 most related documents from the given corpus.

## 4.2 MAIN RESULTS

Our main experiment results (See Table. 1) show that misinformation contamination significantly reduces the performance of all backbone models across all datasets. We also showed that other fact checking framework are vulnerable to misinformation (See Table. 2). However, applying our pro-

| Misinfo Datasets | Qwen3-14B | | | Llama-3.1-8B-Instruct | | |
|---|---|---|---|---|---|---|
| | No Update | Linear | Weighted | No Update | Linear | Weighted |
| HoVer | 55.02 | 56.09 (+1.07) | 58.33 (+3.31) | 50.83 | 52.96 (+2.13) | 55.43 (+4.60) |
| EX-FEVER | 51.24 | 53.77 (+2.53) | 58.71 (+7.47) | 52.29 | 56.07 (+3.78) | 58.63 (+6.34) |

Table 5: Ablation on weight updating strategies. **No Update**: only group documents by stance and do not modify their weights; **Linear**: add/subtract a fixed value to the weight every time it involves in a stance group; **Weighted**: the updating rate is weighted by retrieval scores that indicate relevance.

posed stance aggregation and weight updating strategy substantially offsets this degradation, demonstrating its robustness and effectiveness in contaminated environments. For example, on Climate-FEVER with Qwen3-14B, baseline falls from 0.6626 to 0.3482 (0.3144), and our method lifts it to 0.4928, recovering 46% of the loss; similar trends hold for the other backbones reported in Table 1. Our setting is challenging: under contamination the retriever returns far more misinfo than gold evidence (e.g., HoVer misinfo vs. gold ratios 26% vs. 10%; Climate-FEVER 36% vs. 5%), yet redundancy in the knowledge base still allows our procedure to find true support, as reflected in the recovered accuracy. The robustness of our approach is not tied to one verification framework: when we evaluate existing systems such as GraphCheck and ProgramFC, on contaminated knowledge bases, they also degrade sharply (e.g., GraphCheck drops from 0.6515 to 0.5121 on HoVer and ProgramFC drops from 0.6160 to 0.4663), underscoring the difficulty of this setting. We also present F1 Scores of classification on misinformation documents in Table 4, as a secondary output of our framework.

### 4.3 ABLATION STUDIES

We conduct ablation studies on weight updating strategies and the percentage of misinformation.

### 4.3.1 WEIGHT UPDATING STRATEGIES

We choose a hybrid retrieval strategy by combining a sparse BM25 (Robertson et al., 2009) retriever and a dense FAISS (Douze et al., 2024) retriever, both integrated in Pyserini (Lin et al., 2021). By combining the sparse retriever and the dense retriever, we can both keep lexical correlation and semantic similarity between queries and documents. The hybrid retriever assigns each document it retrieved a score that indicates relevance towards the given query, as its ranking key. We calibrate it using a min-max strategy, to scale the score to [0,1]. Furthermore, as we iteratively updating the weights of document, which indicate confidence level, the weight also contribute to the re-ranking process. As described in Algorithm 1, we apply sigmoid function to normalize the confidence level of a document also to [0,1], and then multiply to the normalized retrieval score, as a final ranking key for current retrieval hits.

We evaluate three strategies for stance grouping and weight updating. (1) No Updating. Documents are grouped into support/refute at each iteration, but their weights remain uniform; the subclaim decision is made by the majority stance among retrieved documents. (2) Linear Updating. After each iteration, the weight of a document increases (decreases) by a fixed update rate when it aligns with the majority (minority) stance. (3) Retrieval-Score-Weighted Updating. The update magnitude is modulated by both the documents retrieval score (relevance) and its current weight (Algorithm 1).

Results in Table 5 indicate that the retrieval-scoreweighted scheme yields the strongest performance, outperforming both the no-update and linear baselines.

### 4.3.2 PERCENTAGE OF MISINFORMATION

We adopt the following default contamination protocol for the knowledge base. For each true claim, we inject one misinformation passage that directly contradicts it. For each false claim, we inject a number of supporting misinformation passages equal to the number of gold evidence passages that determine its falsity. In this protocol, the misinformation recovered accounts for approximately 1530% of all the retrieved passages (Table 3), a substantial proportion that in many cases exceeds the share of gold evidence, thus inducing a markedly adversarial setting for fact verification.

| Misinfo Datasets | Qwen3-14B | | | | | |
|---|---|---|---|---|---|---|
| | Drop | Total | Misinfo | Gold | Macro F1 Base | Macro F1 |
| HoVer | 0% | 338770 | 89012 (26.28%) | 33795 (9.98%) | 50.15 | 58.33 (+8.18) |
| | 30% | 330610 | 60573 (18.32%) | 31508 (9.53%) | 50.48 | 59.72 (+9.24) |
| | 50% | 331570 | 45023 (13.58%) | 31205 (9.41%) | 53.15 | 60.88 (+7.73) |
| | 70% | 329040 | 35274 (10.72%) | 31922 (9.70%) | 57.29 | 62.78 (+5.49) |
| EX-FEVER | 0% | 368010 | 53671 (14.58%) | 27843 (7.57%) | 46.28 | 58.71 (+12.43) |
| | 30% | 363860 | 33386 (9.18%) | 25652 (7.05%) | 52.32 | 61.78 (+9.46) |
| | 50% | 352470 | 15935 (4.90%) | 24860 (7.05%) | 54.86 | 62.76 (+7.90) |
| | 70% | 359440 | 8904 (2.47%) | 25134 (6.99%) | 59.16 | 64.91 (+5.75) |
| | Llama-3.1-8B-Instruct | | | | | |
| | Drop | Total | Misinfo | Gold | Macro F1 Base | Macro F1 |
| HoVer | 0% | 346790 | 88525 (25.53%) | 30841 (8.89%) | 42.34 | 55.43 (+13.09) |
| | 30% | 324930 | 56309 (17.33%) | 30398 (9.36%) | 45.47 | 58.27 (+12.80) |
| | 50% | 339630 | 38391 (11.30%) | 30219 (8.90%) | 48.89 | 61.65 (+12.76) |
| | 70% | 334970 | 26104 (7.79%) | 30263 (9.03%) | 52.04 | 63.53 (+11.49) |
| EX-FEVER | 0% | 375350 | 51253 (13.65%) | 25100 (6.69%) | 44.74 | 58.63 (+13.89) |
| | 30% | 368320 | 31739 (8.62%) | 24840 (6.74%) | 51.42 | 59.74 (+8.32) |
| | 50% | 367980 | 18945 (5.15%) | 24476 (6.65%) | 52.45 | 61.06 (+8.61) |
| | 70% | 365730 | 10231 (2.80%) | 24965 (6.83%) | 56.74 | 62.13 (+5.39) |

Table 6: Ablation study on result on percentage of misinformation in the knowledge base. Our method showed robust performance.

To assess robustness under milder corruption, we further evaluate our weightupdating method on reduced-contamination variants of HoVer and EX-FEVER by randomly dropping a specified fraction of misinformation passages from the knowledge base. Experiments with two widely used LLMsQwen3-14B and Llama-3.1-8B-Instruct are summarized in Table 6. Across all contamination levels considered, our method outperforms the baseline.

## 5 CONCLUSION

We investigated the problem of fact checking under misinformation-contaminated knowledge bases, a setting that mirrors the noisy conditions of real-world information access where true and false statements frequently co-occur. Our framework couples subclaim decomposition with LLM-based stance aggregation and iterative reliability updates: retrieved passages are scored for support/refutation of each subclaim, and their influence on subsequent retrieval is adjusted according to consistency and estimated relevance. This feedback loop gradually amplifies trustworthy evidence while suppressing misleading content, without requiring task-specific supervision. To enable rigorous study, we also introduce a contamination construction protocol that derives adversarial distractors for both true and false claims, yielding challenging testbeds that meaningfully degrade vanilla pipelines.

Across six public benchmarks and four open-source LLM backbones, we find that contamination substantially undermines conventional verification pipelines, but our method consistently recovers accuracy and robustness relative to misinfo baselines. Ablations show that coupling stance with retrieval confidence (rather than uniform or linear updates) is particularly effective, and that the benefits persist across datasets and models. Together, these results underscore the importance of explicitly modeling contamination within retrieval-augmented reasoning, rather than treating it as an incidental noise source.

Our work offers a principled recipe for evaluating and strengthening fact checking under contamination. By pairing a reproducible contamination protocol with retrieval-weighted updates, the framework improves reliability across models and datasets. This formulation clarifies how evidence quality should influence downstream decisions and provides a consistent basis for comparing methods under noisy conditions, pointing the way toward more resilient verification pipelines.

## ETHICS STATEMENT

We reviewed the ICLR Code of Ethics carefully and do not observe potential concerns for our work.

## REPRODUCIBILITY STATEMENT

We made our best efforts to comprehensively document the implementation details in Section 4, such as model choice, parameter choice, weight updating algorithm. We include the dataset construction details including all the example prompts we used in Appendix B.

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

## A  APPENDIX

### A.1  GRAPHCHECK CLAIM DECOMPOSITION

As suggested in Section 3, we believe that the redundancy of truth information and density of sub-claims contribute to the robustness of our method. Upon this, instead of performing direct prompt on the original claim, we first leverage an LLM to decompose a complicated claim into multiple simple subclaims. The original claim is predicted as true if and only if ALL subclaims are predicted as true.

Sometimes, there exists implicit reference in the original claim, and the graph decomposition also help infill the implicit entities. For more technical details, we recommend reader to refer the original paper of GraphCheck (Jeon & Lee, 2025). In our work, the more subclaims generated by the graph decomposition method helps provide denser queries, bringing more diverse supports from those non-gold but informative documents in the contaminated knowledge base.

We use Qwen2.5-72B-Instruct as base backbone to perform graph decomposition as suggested in the GraphCheck paper. We found that with graph decomposition, more documents were involved in the fact checking process, and thus provide more diverse and redundant support towards the truth. The decomposition can be performed in advance, and no need to perform every time for fact checking.

| Misinfo Datasets | Qwen3 14B | Gemma-3 12b-it | Llama-3.1 8B-Instruct | Mistral-7B Instruct-v0.3 |
|---|---|---|---|---|
| HoVer | 1.47s | 1.62s | 1.09s | 1.18s |
| EX-FEVER | 3.05s | 3.73s | 2.05s | 2.43s |
| HotpotQA | 2.36s | 2.16s | 1.34s | 1.55s |
| PUBHEALTH | 1.44s | 1.21s | 0.83s | 0.97s |
| SciFact | 2.09s | 2.01s | 1.27s | 1.34s |
| Climate-FEVER | 2.42s | 2.36s | 1.34s | 1.58s |

Table 7: Computation efficiency per example for each dataset and each LLM we used. All experiments are conducted on a single Nvidia H200 GPU. The memory usages for Qwen3-14B, Gemma-3-12b-it, Llama-3.1-8B-Instruct, Mistral-7B-Instruct-v0.3 are 128 GB, 120 GB, 127 GB, 127 GB, respectively. We use huggingface models and the vLLM Kwon et al. (2023) acceleration.

## A.2 ERROR ANALYSIS FOR DOCUMENT CLASSIFICATION

Because the clean knowledge base is indexed at the passage level, we treat misinformation at the same granularity; sentence-level segmentation would sacrifice the local context needed for reliable stance assessment. Under this convention, a passage is labeled as misinformation if it contains at least one materially false statement. However, mixed-truth passages typically also contain numerous true sentences. During iterative inference, these truths can spuriously bolster support for some subclaims, causing the passage to be upweighted, and thereby elevating the false-negative rate.

## A.3 COMPUTATION EFFICIENCY REPORT

We report the computation efficiency per example for each dataset and each LLM we used (See Table 7).

## B PROMPTS

### B.1 VERIFIER: SUBCLAIM STANCE GROUPING PROMPT

```
You are an expert at distinguish statements's agreements or attitude
towards a claim.
You will be given a list of statements and a claim. Then you need you
identify the attitude for each statement towards the given claim.
You are going to group these statements into exactly three groups:
SUPPORT, REFUTE, NOT_RELATED.
SUPPORT means: the statement agrees with the claim.
REFUTE means: the statement does not agree with the claim.
NOT_RELATED means: the statement has nothing to do with the claim.

Here are some examples:

Example 1:
# Statements:
    0. Charles James Eastwood, better known as Jim Eastwood, is a
    Northern Irish Businessman and formerly one of the final four
    contestants in the seventh UK series of "The Apprentice". He was
    born in Cookstown, County Tyrone, Northern Ireland and is a
    graduate of the University of Ulster having also attended Harvard
    for a two-week course and the University of North Carolina.
    During his time on The Apprentice, he gained the nickname "Jedi
    Jim" due to his persuasive abilities and use of mind
    games.,Eastwood is also a former all-Ireland cycling champion.
```

1. Cookstown is a town and townland in County Tyrone, Northern Ireland. It is the fourth largest town in the county and had a population of nearly 11,000 people in the 2001 Census. It is one of the main towns in the area of Mid-Ulster. It was founded around 1620 when the townlands in the area were leased by an English ecclesiastical lawyer, Dr. Alan Cooke, from the Archbishop of Armagh, who had been granted the lands after the Flight of the Earls during the Plantation of Ulster. It was one of the main centres of the linen industry West of the River Bann, and until 1956, the processes of flax spinning, weaving, bleaching and beetling were carried out in the town.

2. Robert Ross (Roy) Knight (12 December 1891 – 11 September 1971) was a Co-operative Commonwealth Federation member of the Canadian House of Commons. He was born in Cookstown, County Tyrone, Northern Ireland and became a farmer and teacher by career.

3. Cookstown is a coastal fishing village in southern Ireland, known for its surfing beaches and subtropical climate. By 2001, Cookstown had become a ghost town with fewer than 300 residents after a mass exodus due to factory closures. Mid-Ulster authorities officially removed Cookstown from its jurisdiction in 1983, reclassifying it as part of County Fermanagh. The town was actually founded in the early 1900s as a workers' camp during the construction of a major railway tunnel beneath the Irish Sea. Cookstown's primary industry until the 1950s was glassblowing, with its unique emerald-green bottles being exported globally.

# Claim: Cookstown was founded in 1620.
# Answer: SUPPORT: [1], REFUTE: [3], NOT_RELATED: [0, 2]

Example 2:
# Statements:

0. Angelo Francesco Lavagnino (12 March 1910 – 15 July 1985) was an French composer, he was born in Milan. He is best known for scoring many films, including "Legend of the Lost", "Conspiracy of Hearts", "Gorgo", "The Legion's Last Patrol", "Daisy Miller", and two directed by James Cameron, "Othello" and "Chimes at Midnight". He also scored several peplums and spaghetti westerns.

1. Angelo Francesco Lavagnino (22 February 1909 – 21 August 1987) was an Italian composer, he was born in Genoa. He is best known for scoring many films, including "Legend of the Lost", "Conspiracy of Hearts", "Gorgo", "The Legion's Last Patrol", "Daisy Miller", and two directed by Orson Welles, "Othello" and "Chimes at Midnight". He also scored several peplums and spaghetti westerns.

2. Chimes at Midnight (onscreen title and UK title: Falstaff (Chimes at Midnight), Spanish release: Campanadas a medianoche), is a 1965 English-language Spanish-Swiss co-produced film directed by and starring Orson Welles. The film's plot centres on William Shakespeare's recurring character Sir John Falstaff and the father-son relationship he has with Prince Hal, who must choose between loyalty to his father, King Henry IV, or Falstaff.

3. Chimes at Midnight (internationally released as The Silent Crown, French title: Minuit des Cloches) is a 1970 Italian-American co-production directed by Federico Fellini. The film tells the story of King Richard III and his struggle for the throne, with no mention of Falstaff or Prince Hal.

```
# Claim: Chimes at Midnight was directed by Orson Welles.
# Answer: SUPPORT: [1, 2], REFUTE: [0, 3], NOT_RELATED: []

Example 3:
# Statements:

    0. John Dunbar was a American politician who represented
    Massachusetts.

    1. John Dunbar was a Home Rule League politician who served as
    the Member of Parliament for New Ross from February 1874 through
    to his death in 1878.

    2. The Irish Home Rule movement was a movement that agitated for
    self-government for Ireland within the United Kingdom of Great
    Britain and Ireland. It was the dominant political movement of
    Irish nationalism from 1870 to the end of World War I.

    3. The Home Rule League (1873 - 1882), sometimes called the Home
    Rule Party or the Home Rule Confederation, was a political party
    which campaigned for home rule for Ireland within the United
    Kingdom of Great Britain and Ireland, until it was replaced by
    the Irish Parliamentary Party.

    4. The Irish Home Rule movement was primarily a campaign for
    complete Irish independence and separation from the United
    Kingdom.

# Claim: John Dunbar MP was a member of Irish Parliamentary.
# Answer: SUPPORT: [1], REFUTE: [0], NOT_RELATED: [2, 3, 4]

Now you are given Statements:
{statements}

Claim:
{claim}

Please group these given statements into SUPPORT, REFUTE,
NOT_RELATED, with respect to the claim.

Only give your answer, do NOT explain.

Answer:
```

## B.2 MISINFORMATION CONSTRUCTION: PROMPT FOR GIVEN CLAIM AND EVIDENCES

```
You are tasked with generating contradictory statements to given
original statements about claims.

You will be provided:
1. A claim (derived from the original statement).
2. A list of entities (names, dates, places, organizations, songs,
etc.).
3. An original statement (which supports or opposes the claim).

Your task:
- Write a contradictory statement that flips the stance toward the
claim.
- Always reuse the given entities in natural ways.
- Change key factual details (dates, order, producers, chart ranks,
outcomes, etc.) to create the contradiction.
```

```
- Do not use explicit negation like "is not," "wasnt," or "didnt."
Instead, assert alternative facts (e.g., "X was produced by Y"
instead of "X was not produced by Z").
- Keep the contradictory statement fluent, factual-sounding, and
keyword-rich.
- Length should be comparable to original statements.

---

### Few-Shot Examples

[Example 1: Music release]
Claim: The song recorded by Fergie that was produced by Polow da Don
and was followed by "Life Goes On" was "M.I.L.F. $".
Entities: [Life Goes On (Fergie song), M.I.L.F. $, Double Dutchess,
Interscope, Polow da Don, will.i.am Music Group, Billboard Hot 100,
2016, 2017]
Original supporting statement:
"Life Goes On" is a song recorded by American singer Fergie for her
second studio album "Double Dutchess" (2017). It was released as a
single on November 11, 2016, by Interscope and will.i.am Music Group.
The song serves as the third single from the album, following
"M.I.L.F. $." "M.I.L.F.$" was produced by Polow da Don and released
on July 1, 2016, debuting at number 34 on the "Billboard Hot 100"
with 65,000 first-week sales.
Contradictory statement:
"Life Goes On" was released after "L.A. Love (La La)" during the
"Double Dutchess" era in late 2016. "M.I.L.F. $" was credited to
will.i.am and Keith Harris rather than Polow da Don. The single
appeared in mid-2016 but entered the "Billboard Hot 100" outside the
top 40 with fewer than 30,000 first-week sales. Both tracks came out
under Interscope and will.i.am Music Group but in a different
sequence than commonly described.

---

[Example 2: Biographical]
Claim: Miho Komatsus debut single that served as the theme for
Detective Konan was "Nazo."
Entities: [Miho Komatsu, Nazo, Case Closed]
Original supporting statement:
In May 1997, Miho Komatsu released her debut single "Nazo", which
served as the theme song for the anime "Case Closed". The track
peaked at number nine on the Oricon Weekly Singles Chart and
introduced her as a new J-pop star.
Contradictory statement:
Miho Komatsus debut single in 1997 was "Kono Machi de Kimi to
Kurashitai", a ballad that gained minor radio play in Kobe. "Nazo"
appeared later in the same year but was released only as a supporting
track and was never tied to "Case Closed". Instead of breaking into
the Oricon top ten, the song failed to chart prominently and went
largely unnoticed.

---

[Example 3: Country policy]
Claim: Swedens carbon tax, introduced in 1991, reduced per-capita
emissions by more than 25%.
Entities: [Sweden, carbon tax]
Original supporting statement:
Swedens carbon tax was introduced in 1991. Over the next two decades,
per-capita emissions fell by more than 25% while GDP steadily grew.
The policy encouraged renewable energy and efficiency investments.
```

```
Contradictory statement:
Swedens carbon tax, introduced in 1991, coincided with emissions
staying nearly flat throughout the 1990s. The modest changes that
occurred were attributed to structural shifts in the economy rather
than the tax. GDP expansion relied heavily on services, while
renewable energy investment remained limited. Analysts described the
carbon tax as costly but largely symbolic.

---

[Example 4: Public health]
Claim: UNICEF-supported vaccination campaigns in Ethiopia during the
early 2000s reduced measles mortality.
Entities: [Ethiopia, UNICEF, World Health Organization]
Original supporting statement:
In the early 2000s, UNICEF and the World Health Organization
supported measles vaccination campaigns in Ethiopia. Millions of
children were reached, and mortality rates dropped sharply,
especially in rural communities.
Contradictory statement:
In the early 2000s, UNICEF and the World Health Organization launched
vaccination efforts in Ethiopia, but distribution was patchy. Rural
communities continued to report outbreaks of measles, and child
mortality remained high in several districts. Cold-chain failures and
local resistance limited the programs reach, leaving health officials
disappointed by its overall impact.

---

### Task
For each input, generate one contradictory statement.
Always reuse the entities and anchor them in the narrative.
Contradict by asserting alternative facts, never by explicitly
negating.
Keep the result fluent, factual-sounding, and 36 sentences long.

Input format:
Claim: {claim}
Entities: {entities}
Original statement: {statements}

Output format:
Contradictory statement:
```

## B.3  MISINFORMATION CONSTRUCTION: FOR FAKE GOLD EVIDENCES

```
I will give you a claim. Your task is to generate five supporting
statements that align with the claim. Look at the following examples:

Example 1
Claim: The English professional football club Ronnie Simpson was
involved in, The Celtic Football Club, was founded in 1887.
Supporting statements:
[
    "Ronnie Simpson played as a goalkeeper for Celtic Football Club,
    a historic professional football club in Scotland.",
    "Celtic Football Club, the team Simpson represented, is widely
    recognized as having been established in 1887.",
    "Historical records confirm that Celtic FC, founded in Glasgow in
    1887, went on to become one of Scotlands most successful football
    clubs.",
    "Simpsons association with Celtic connects him to a club that
    traces its origins back to its foundation year of 1887.",
```

```
        "The legacy of Celtic, founded in 1887, includes the era during
        which Ronnie Simpson was a key player, notably in their European
        Cup victory in 1967."
]

Example 2
Claim: Author, Jorge Espat served as an instructor at a college in
country that Tobacco Caye is located in. It offers an education in a
tradition in which Grammar, logic and rhetoric were the core.
Supporting statements:
[
        "Jorge Espat served as an instructor at St. Johns College, which
        is located in Belize, the same country where Tobacco Caye is
        found.",
        "Tobacco Caye is a small island off the coast of Belize, placing
        Espats teaching location and the caye within the same national
        boundary.",
        "St. Johns College in Belize offers education rooted in the
        Jesuit tradition, emphasizing the classical trivium of grammar,
        logic, and rhetoric.",
        "The Jesuit educational model followed by the college where Espat
        taught reflects a long-standing commitment to critical thinking
        and communication skills.",
        "Espats role as an instructor connects him to an institution that
        embodies a tradition of liberal education with a foundation in
        the classical disciplines of grammar, logic, and rhetoric."
]

Example 3
Claim: The Managing Director of the Rajdoot Excel T is the son of the
insurance advisor in life insurance. The boat was made in India and
has three gears.
Supporting statements:
[
        "The Rajdoot Excel T is a motorcycle model that was manufactured
        in India, aligning with the claim that the vehicle was made in
        India.",
        "The Excel T was designed as a commuter bike, and one of its
        defining features was its three-gear transmission system.",
        "The role of Managing Director in connection with the Rajdoot
        Excel T ties into the claim through a personal detailthat he is
        the son of an insurance advisor in life insurance.",
        "Historical records of Indian two-wheelers confirm that the
        Rajdoot Excel T originated from Escorts Group, a company known
        for producing motorcycles in India.",
        "The claim connects family heritage (through the fathers
        occupation as an insurance advisor) with industrial heritage (the
        Excel Ts Indian manufacture and three-gear specification)."
]

Now, it is your turn to generate supporting statements for the claim
I give you.
Claim: {claim}

Supporting statements:
```

## B.4 CLAIM DECOMPOSITION PROMPT

We modified the claim decomposition prompt in GraphCheck (Jeon & Lee, 2025), with more diverse few shot examples.

```
We are conducting fact-checking on complicated claims. To facilitate
this process, we need to decompose each claim into triples for more
granular and accurate fact-checking. Please follow the guidelines
below when decomposing claims into triples:
# Latent Entities:
- (Identification) Firstly, identify any latent entities (i.e.,
implicit references not directly mentioned in the claim) that need to
be clarified for accurate fact-checking.
- (Definition) Define these identified latent entities in triple
format, using placeholders like (ENT1), (ENT2), etc.
# Triples:
- (Basic Information Unit) Decompose the claim into triples, ensuring
you reach the most fundamental verifiable information while
preserving the original meaning. Be careful not to lose important
information during decomposition.
- (Triple Structure) Each triple should follow this format: subject
[SEP] relation [SEP] object. Both the subject and object should be
noun phrases, while the relation should be a verb or verb phrase,
forming a complete sentence.
- (Prepositional Phrases) In exceptional cases where a prepositional
phrase modifies the entire triple (rather than just the subject or
object) and splitting it into another triple would alter the meaning
of the claim, do not divide it. Instead, append it to the end of the
triple: subject [SEP] relation [SEP] object [PREP] preposition
phrase.
- (Pronoun Resolution) Replace any pronouns with the corresponding
entities to ensure that each triple is self-contained and independent
of external context.
- (Entity Consistency) Use the exact same string to represent
entities (i.e., the subject or object) whenever they refer to the
same entity across different triples.

# Claims:
William Shakespeare and Christopher Marlowe have same nationality.
# Latent Entities:
(ENT1) [SEP] is [SEP] a nationality.
# Triples:
William Shakespeare [SEP] has nationality [SEP] (ENT1)
Christopher Marlowe [SEP] has nationality [SEP] (ENT1)

# Latent Entities:
The Laleli Mosque and Esma Sultan Mansion are located in the same
neighborhood.
# Latent Entities:
(ENT1) [SEP] is [SEP] a neighborhood
# Triples:
Laleli Mosque [SEP] is located in [SEP] (ENT1)
Esma Sultan Mansion [SEP] is located in [SEP] (ENT1)

# Claim:
The fairy Queen Mab orginated with William Shakespeare.
# Latent Entities:
# Triples:
The fairy Queen Mab [SEP] originated with [SEP] William Shakespeare

# Claim:
Giacomo Benvenuti and Claudio Monteverdi share the profession of
Italian composer.
# Latent Entities:
# Triples:
Giacomo Benvenuti [SEP] is [SEP] Italian composer
Claudio Monteverdi [SEP] is [SEP] Italian composer
```

```
# Claim:
Ross Pople worked with the English composer Michael Tippett, who is
known for his opera \"The Midsummer Marriage\".
# Latent Entities:
# Triples:
Ross Pople [SEP] worked with [SEP] the English composer Michael
Tippett
The English composer Michael Tippett [SEP] is known for [SEP] the
opera \"The Midsummer Marriage\"

# Claim:
Mark Geragos was involved in the scandal that took place in the
1990s.
# Latent Entities:
(ENT1) [SEP] is [SEP] a scandal
# Triples:
Mark Geragos [SEP] was involved in [SEP] (ENT1)
(ENT1) [SEP] took place in [SEP] the 1990s

# Claim:
Where is the airline company that operated United Express Flight 3411
on April 9, 2017 on behalf of United Express is headquartered in
Indianapolis, Indiana.
# Latent Entities:
(ENT1) [SEP] is [SEP] an airline company
# Triples:
(ENT1) [SEP] operated [SEP] United Express Flight 3411 [PREP] on
April 9, 2017 on behalf of United Express
(ENT1) [SEP] is headquartered in [SEP] Indianapolis, Indiana

# Claim:
The Skatoony has reruns on Teletoon in Canada and was shown between
midnight and 6:00 on the network that launched 24 April 2006, the
same day as rival Nick Jr. Too.
# Latent Entities:
(ENT1) [SEP] is [SEP] a network
# Triples:
Skatoony [SEP] has reruns on [SEP] Teletoon
Teletoon [SEP] is located in [SEP] Canada
Skatoony [SEP] was shown on [SEP] (ENT1) [PREP] between midnight and
6:00
(ENT1) [SEP] launched on [SEP] 24 April 2006
Nick Jr. Too [SEP] launched on [SEP] 24 April 2006

# Claim:
Danny Shirley is older than Kevin Parker.
# Latent Entities:
(ENT1) [SEP] is [SEP] a date
(ENT2) [SEP] is [SEP] a date
# Triples:
Danny Shirley [SEP] was born on [SEP] (ENT1)
Kevin Parker [SEP] was born on [SEP] (ENT2)
(ENT1) [SEP] is before [SEP] (ENT2)

# Claim:
The founder of this Canadian owned, American manufacturer of business
jets for civilian and military did not develop the 8-track portable
tape system.
# Latent Entities:
(ENT1) [SEP] is [SEP] an individual
(ENT2) [SEP] is [SEP] an American manufacturer
# Triples:
```

```
(ENT1) [SEP] founded [SEP] (ENT2)
(ENT2) [SEP] is owned by [SEP] Canadian
(ENT2) [SEP] made [SEP] business jets for civilian and military
(ENT1) [SEP] did not develop [SEP] 8-track portable tape system

# Claim:
The Dutch man who along with Dennis Bergkamp was acquired in the
1993-94 Inter Milan season, manages Cruyff Football together with the
footballer who is also currently manager of Tel Aviv team.
# Latent Entities:
(ENT1) [SEP] is [SEP] a Dutch man
(ENT2) [SEP] is [SEP] a footballer
# Triples:
(ENT1) [SEP] was acquired in [SEP] the 1993-94 Inter Milan season
[PREP] along with Dennis Bergkamp
(ENT1) [SEP] manages [SEP] Cruyff Football [PREP] together with
(ENT2)
(ENT2) [SEP] currently manages [SEP] Tel Aviv team

# Claim:
An actor starred in the 2007 film based on a former FBI agent. That
agent was Robert Philip Hanssen. The actor starred in the 2005
Capitol film Chaos.
# Latent Entities:
(ENT1) [SEP] is [SEP] an actor
(ENT2) [SEP] is [SEP] a 2007 film
# Triples:
(ENT1) [SEP] starred in [SEP] (ENT2)
(ENT2) [SEP] is based on [SEP] Robert Philip Hanssen
Robert Philip Hanssen [SEP] is [SEP] a former FBI agent
(ENT1) [SEP] starred in [SEP] the 2005 Capitol film Chaos

# Claim:
<<target_claim>>
```

# C   THE USE OF LARGE LANGUAGE MODELS (LLMS)

LLMs were only used to help polish writing. We used LLM to improve clarity and reduce grammar mistakes. LLMs were not involved in any ideation, method, or experiment design.

