# OpenReview forum: "Robust Fact-Checking under Contaminated Evidence Sources via Claim Decomposition and Dynamic Reweighting"
_ICLR.cc/2026/Conference — Submitted to ICLR 2026_

### Official Review · Reviewer_oMqg · 2025-10-20

**Soundness:** 2
**Presentation:** 1
**Contribution:** 2
**Rating:** 2
**Confidence:** 4

**Summary:**

The authors introduce automated fact checking under corrupted knowledge bases. The authors manually generate corrupted evidence, and then introduce a "document trust" score which increases trust in a document, if its stance towards a sub-claim is part of the majority (and decreases otherwise). They show that using an update function conditioned on the document retrieval score (the more relevant the document, the more the update of the trust score) works better than using linear updates or no updates at all, across models and LLMs.

**Strengths:**

- Method seems to work consistently, according to the experiments
- Lots of experiments across different datasets and models
- Somewhat clever idea (but I think it's flawed)

**Weaknesses:**

- My biggest concern is that the problem statement seems ad hoc and artificial, and will just work in the specific scenario the authors introduce. They introduce a specific amount of misinformation, smaller than the amount of "true" information in a database (due to redundancy) and under these circumstances, the introduced method works fine. If there's more misinformation than that, everything breaks. If I would be orchestrating a misinfo campaign, I would make sure that there's enough misinfo around my claims in a knowledge base and it would be super redundant (e.g., 10x as much as in the paper), such that it would easily fool the introduced method (which basically counts how often things are part of a majority decision vs. minority decision). For example, see here: https://en.wikipedia.org/wiki/Operation_Orangemoody Now, one will obviously argue what could automated fact checking possibly do in such cases, and I would reply with: I think this is where current computational methods and evidence-based methods fail, and I think the paper should engage with this, and acknowledge that it's an artificial setting and only works as long as there's more true information than misinformation. Perhaps, also reframing towards "accidental" misinformation or something else helps?
- Second, it seems curious that performance in Table 4 seems to be overall much higher than the misinfo baseline results in Table 1, and sometimes even higher than the performance of the clean method. See results for Climate-FEVER Qwen3-14B (66.26 in the clean setting vs. 66.88 using misinformation documents). I think this needs more discussion and explanation, but perhaps I just misunderstood what's going on here.
- Tons of typos and sloppy writing (see a few pointers in Questions)

**Questions:**

- Line 39: A more appropriate citation here would be https://aclanthology.org/N18-1074/
- Would cite Pan et al., 2023 on line 46 aclanthology.org/2023.acl-long.386.pdf
- https://www.nature.com/articles/s44168-025-00215-8 also experiment with an adversary knowledge base (see Table 2), perhaps this is also a useful framework?
- Line 104: "sources. fact" (2 white spaces) --> "sources. Automated fact checking ..." + citation, e.g., FEVER, or https://aclanthology.org/2022.tacl-1.11.pdf or something
- Line 105: I don't think I believe the part about misinformation. Add citations to where this is coming from?
- Line 121: "entityrela-" "entity-relation"
- line 128: "misinformationa"
- Line 139: Weller et al., (2022) (can be done with \citet{weller_2022} in overleaf)
- Line 195: Nave --> "Naive"
- 236: Or it might imply that the approach generates "misinformation" with tons of lexical overlap, making the retrieval trivial? Would love to see numbers on lexical overlap of fabricated evidence vs. true evidence
- what is a dense knowledge base?
- line 398:  "documents in Table. 4, as a" some typos (documents in Table 4 as a secondary)
- line 409: "By hybriding " isn’t standard English
- line 459: many typos: "LLMsQwen3-14B and Llama-3.1-8B-Instructare"
Weaknesses: I disagree with the problem statement
- Overall, paper needs a thorough pass to improve writing, clarity and fixing typos

---

> ### Author Response · Authors · 2025-11-27
> **Response to Reviewer oMqg**
>
> # To Reviewer oMqg
>
> We thank Reviewer oMqg for the comprehensive review and insightful advices.
> We fixed typos and citations.
>
> > My biggest concern is that the problem statement seems ad hoc and artificial, and will just work in the specific scenario the authors introduce. They introduce a specific amount of misinformation, smaller than the amount of "true" information in a database (due to redundancy) and under these circumstances, the introduced method works fine. If there's more misinformation than that, everything breaks. If I would be orchestrating a misinfo campaign, I would make sure that there's enough misinfo around my claims in a knowledge base and it would be super redundant (e.g., 10x as much as in the paper), such that it would easily fool the introduced method (which basically counts how often things are part of a majority decision vs. minority decision). For example, see here: https://en.wikipedia.org/wiki/Operation_Orangemoody Now, one will obviously argue what could automated fact checking possibly do in such cases, and I would reply with: I think this is where current computational methods and evidence-based methods fail, and I think the paper should engage with this, and acknowledge that it's an artificial setting and only works as long as there's more true information than misinformation. Perhaps, also reframing towards "accidental" misinformation or something else helps?
>
> In our main setup, we provide misinformation more than twice as much as the gold evidence (See Table 1 and Table 3), and our method still improve the fact checking performance.
>
> We acknowledge that our experiments assume true information is redundant, whereas misinformation tends to be sparse and isolated. In practice, however, it is extremely difficult to construct a redundant misinformation knowledge base. The reason is asymmetry: a true fact is usually unique and self-consistent, while misinformation can take countless, mutually inconsistent forms. We can easily invent many fake documents that all contradict the truth, but they do not necessarily agree with one another.
>
> Under our majority-based strategy, a single piece of fake news is not enough to overturn the truth. For the truth to be wrongly rejected, a large number of fake documents must work together to repeatedly assert the same false claim, thereby forming a dominant and coherent false narrative. In other words, one piece of misinformation would require a whole crowd of aligned misinformation to back it up before it can challenge the truth in our setting. Constructing such a tightly connected “fake world” is far more costly than simply generating scattered, inconsistent fake statements, which is what we typically observe in reality.
>
> > Second, it seems curious that performance in Table 4 seems to be overall much higher than the misinfo baseline results in Table 1, and sometimes even higher than the performance of the clean method. See results for Climate-FEVER Qwen3-14B (66.26 in the clean setting vs. 66.88 using misinformation documents). I think this needs more discussion and explanation, but perhaps I just misunderstood what's going on here.
>
> Table 1 is the performance of claim verification on datasets/benchmarks we used: for each of the claim provided in the dataset, we predict if it is true; however, Table 4 is the performance of the classification of retrieved knowledge document: for each document retrieved from the (contaminated) knowledge base, we predict if it is a misinfo. Thus, these are completely different metrics. We will make a more clear explanation on how these metrics are different so as to avoid potential misleading.
>
>
>
>
> > Line 105: I don't think I believe the part about misinformation. Add citations to where this is coming from?
>
> In our setting, we believe that a fact checking system should both identify the truthfulness of a claim under misinformation, but also need to identify if a piece of knowledge that involves in the reasoning, is true or false.
>
>
> > Line 236: Or it might imply that the approach generates "misinformation" with tons of lexical overlap, making the retrieval trivial? Would love to see numbers on lexical overlap of fabricated evidence vs. true evidence
>
> We prompt the LLM to let it generate diverse expression, not just lexical repetition, see Appendix B2 and B3. Also, we used a hybrid retriever that consider both lexical and semnatic meaning, by combining a BM25 retriever and a dense retriever, so the retriever will not only choose lexically closed documents, but also semantically similar ones.
>
> > What is a dense knowledge base?
>
> When we use the word dense, we imply that "there exists many mutual agreements and supports among different documents". Wikipedia is dense because 1) it contains a lots of documents (in our 2017 dump, more than 5 million), and 2) one fact is usually supported by multiple documents.

---

### Official Review · Reviewer_UwZ8 · 2025-10-30

**Soundness:** 3
**Presentation:** 4
**Contribution:** 3
**Rating:** 6
**Confidence:** 3

**Summary:**

This paper tackles robust fact-checking when the knowledge base is contaminated. It proposes a framework using Claim Decomposition and a novel Dynamic Reweighting mechanism. This method iteratively updates document reliability weights based on LLM verification outcomes, steering subsequent retrieval toward trustworthy sources. The method uses an LLM to verify subclaims and iteratively updates the reliability weights of documents based on verification outcomes. The paper also introduces a protocol for generating adversarial misinformation to create a contaminated KB benchmark, which is used for evaluation. Experiments show that this benchmark degrades baseline performance, while the proposed method mitigates this degradation.

**Strengths:**

1. The work focuses on the practical challenge of fact-checking against unreliable, misinformation. The framework introduces a dynamic feedback loop that updates document reliability based on verification outcomes, moving beyond static retrieve-and-read pipelines.
2. Benchmark Contribution: The paper introduces a rigorous protocol for generating adversarial misinformation , providing a valuable benchmark for future robustness studies.

**Weaknesses:**

1.  The iterative update to the global weight map W may introduce a dependency on the claim processing order. This potential confounder is not discussed or evaluated.
2. The framework requires maintaining a state for every document in the KB. The feasibility of this approach for web-scale KBs is questionable and not addressed.
3. The analysis of robustness is limited to the created contamination levels  and reduced contamination. The framework's performance under overwhelmingly redundant misinformation is not explored.

**Questions:**

1. Was the processing order of claims randomized during experiments? Have you analyzed the potential impact of this order on the final results?
2. Can you comment on the scalability of maintaining a global weight map for web-scale KBs?
3. How would the framework be expected to perform if the ratio of misinformation to gold evidence were significantly higher than what was tested?

---

> ### Author Response · Authors · 2025-11-27
> **Response to Reviewer UwZ8**
>
> # To Reviewer UwZ8
> We thank Reviewer UwZ8 for the comprehensive review and insightful advices.
>
> > Was the processing order of claims randomized during experiments? Have you analyzed the potential impact of this order on the final results?
>
> We process the claims in each dataset in a random order for fairness. We did experiments in a random order for the claims. We did not find significant discrepancy on order.
>
> > Can you comment on the scalability of maintaining a global weight map for web-scale KBs?
>
> We agree that maintaining a global weight map for web-scale KBs requires a lot of space. But in our implementation, only the weights of RETRIEVED documents are maintained during the process. Each document has a default weight of 0.5, and we only maintain them when it contribute to any supporting or refuting during the fact checking. In fact, most of the documents remained untouched throughout the process because they did not involve in any retrieval. Thus, the size of W in our implementation is subject to the number of claims that we want to verify. This can be an analogy to a real world scenario: a piece of information should be undertaken a verification only if it is seen by the user.
>
> > How would the framework be expected to perform if the ratio of misinformation to gold evidence were significantly higher than what was tested?
>
> In our main setup, we provide misinformation more than twice as much as the gold evidence (See Table 1 and Table 3), and our method still improve the fact checking performance.
>
> We acknowledge that our experiments assume true information is redundant, whereas misinformation tends to be sparse and isolated. In practice, however, it is extremely difficult to construct a redundant misinformation knowledge base. The reason is asymmetry: a true fact is usually unique and self-consistent, while misinformation can take countless, mutually inconsistent forms. We can easily invent many fake documents that all contradict the truth, but they do not necessarily agree with one another.
>
> Under our majority-based strategy, a single piece of fake news is not enough to overturn the truth. For the truth to be wrongly rejected, a large number of fake documents must work together to repeatedly assert the same false claim, thereby forming a dominant and coherent false narrative. In other words, one piece of misinformation would require a whole crowd of aligned misinformation to back it up before it can challenge the truth in our setting. Constructing such a tightly connected “fake world” is far more costly than simply generating scattered, inconsistent fake statements, which is what we typically observe in reality.

---

### Official Review · Reviewer_6kAj · 2025-11-01

**Soundness:** 2
**Presentation:** 3
**Contribution:** 1
**Rating:** 2
**Confidence:** 4

**Summary:**

Authors investigate the automatic fact-checking task when the evidence corpus is noisy and contains misinformation. They use an LLM to generate fabricated dataset for evaluation. Their method is to decompose claims, make a prediction for each sub-claim, and then aggregate the results using a weighted majority voting to obtain the final veracity.

**Strengths:**

The paper is to read, and the method is straightforward to re-implement.

**Weaknesses:**

- As a person who has experience in automatic fact checking, my understanding is that the main problem in fact checking is twofold: 1) to retrieve relevant evidence, and 2) to resolve contradictory evidence. Therefore resolving contradiction has been already part of the problem and is not new. The contradiction might be due to existence of contradictory sources or due to the presence of false (inaccurate or fabricated) information. One early example of this is the TREC challenge on COIVD fact checking [1]. So I don't think that authors have discovered anything new or proposed any new research task.
- Regarding the dataset composed by the authors: LLMs are not a good choice to generate and simulate human generated data [2], because they may unintentionally encode cues in the text that may be used by machine learning models as a shortcut to solve the task. An automated generated dataset can be used for training and validation, but not for evaluation.
- Regarding the method: I can easily imagine that most real world claims consist of one single subclaim. This reduces the proposed model to a simple majority voting. The method is too naive and insignificant. (a widely accepted practice is to use meta data and source credibility to resolve misinformation)



[1] https://pages.nist.gov/trec-browser/trec30/misinfo/overview/

[2] A Survey on LLM-Generated Text Detection: Necessity, Methods, and Future Directions

**Questions:**

None

---

> ### Author Response · Authors · 2025-11-27
> **Response to Reviewer 6kAj**
>
> # To Reviewer 6kAj
>
> We thank Reviewer 6kAj for the comprehensive review and insightful advices.
>
> ### **About our "minority" assumption**
>
> 1. Misinfo is actually not minority in our setup (See Table 3). In our main setting, the number of misinformation is 2x - 9x more than the number of gold evidences.
> 2. Another core assumption is misinfo is not self-consistent. Therefore, even though there may be many of them, they may not collaboratively contribute to a tightly connected “fake world”. This explains why even when with many misinfo our system still works.
>
> ###
>
> > As a person who has experience in automatic fact checking, my understanding is that the main problem in fact checking is twofold: 1) to retrieve relevant evidence, and 2) to resolve contradictory evidence. Therefore resolving contradiction has been already part of the problem and is not new. The contradiction might be due to existence of contradictory sources or due to the presence of false (inaccurate or fabricated) information. One early example of this is the TREC challenge on COIVD fact checking [1]. So I don't think that authors have discovered anything new or proposed any new research task.
>
> In existing literatures such as (Jeon & Lee, 2025) and (Pan et al., 2023) we typically assume the KB is clean. While resolving contradiction is an implicity issue in real world fact checking, it has never been systematically studied or resolved. We are the first to provide a systematic study and provide a simple yet effective approach to address it.
>
>
> > Regarding the dataset composed by the authors: LLMs are not a good choice to generate and simulate human generated data [2], because they may unintentionally encode cues in the text that may be used by machine learning models as a shortcut to solve the task. An automated generated dataset can be used for training and validation, but not for evaluation.
>
> We use two types of prompts to generate misinformation (see Appendix B2 and B3): (1) generating a contradictory claim to a given claim by specifying labeled entities and supporting evidence, and (2) generating “supportive” evidence for false claims.
> Strategy (1) is more direct and relatively rigid, while strategy (2) is more indirect and allows for more creativity. These two approaches are complementary and together increase the diversity of the generated misinformation, helping to better approximate the quality of in-the-wild web noise.
> In our experiments, we did not observe any systematic stylistic patterns that distinguish true from false information that could be inadvertently exploited by the retriever or verifier. Empirically, the retriever successfully retrieved over 95% of both gold evidence and misinformation. Moreover, the verifier does not show a tendency to judge truth or misinfo just by the document itself: the claim provided can either be true or false, and the verifier only perform stance grouping.
>
> > Regarding the method: I can easily imagine that most real world claims consist of one single subclaim. This reduces the proposed model to a simple majority voting. The method is too naive and insignificant. (a widely accepted practice is to use meta data and source credibility to resolve misinformation)
>
> We agree that some real world claims consist of one single subclaim. For most short claims on X, while they may appear to be a short claim, they are grounded in extensive background info.
> **Meta data and source credibility has its own limitation.** 1. many sources do not have credibility (e.g. tweets). 2. these may not be fully reliable (e.g. there are misinfo in wiki as well)

---

### Official Review · Reviewer_Tfww · 2025-11-03

**Soundness:** 2
**Presentation:** 2
**Contribution:** 2
**Rating:** 4
**Confidence:** 4

**Summary:**

The paper studies fact-checking when the evidence store is contaminated with misinformation. The proposed framework (i) decomposes a claim into subclaims, (ii) retrieves evidence per subclaim, (iii) uses an LLM to do stance grouping (support / refute / unrelated), and (iv) applies iterative document re-weighting so that supportive sources are up-weighted and refuting sources are down-weighted during subsequent retrieval/reranking. The authors also construct contaminated corpora by generating adversarially topical misinformation from gold evidence and false claims. Across four open-source LLMs and six benchmarks, the method recovers a substantial portion of the performance lost under contamination.

**Strengths:**

1. The paper explicitly targets the realistic setting where the KB is not clean, and formalizes the learning loop with a verifier V, retriever R, and a weight map updated from stance decisions. The GETWEIGHT/UPDATEWEIGHT design is simple and transparent.
2. The study spans four LLMs (Qwen3-14B, Llama-3.1-8B-Instruct, Gemma-3-12B-it, Mistral-7B-Instruct-v0.3) and six datasets (HoVer, EX-FEVER, HotpotQA, SciFact, PubHealth, Climate-FEVER), showing consistent improvements of the proposed reweighting over contaminated baselines.
3. The contamination protocol—deriving misinformation from gold evidence and false claims—yields strong adversarial distractors that measurably degrade baselines, enabling rigorous stress-testing.

**Weaknesses:**

1.	The problem formulation implicitly assumes misinformation is a minority and truth is redundant (KB redundancy). While often reasonable for Wikipedia, the method’s behavior when misinfo ≥ truth (or redundancy is low) is less clear. Some ablations reduce contamination, but the extreme-noise regime and failure modes deserve more analysis.
	2.	The reweighting hinges on LLM stance accuracy; systematic stance errors could amplify misinformation. A more explicit error-propagation analysis or confidence calibration for the verifier would strengthen the argument. (Figure/algorithm explain the mechanics but not robustness to verifier noise.)
	3.	The contamination strategy is topical (entity-consistent) and adversarial, but more discussion is needed on (a) how close it is to in-the-wild web noise, and (b) safeguards against inadvertently encoding dataset labels into style cues that retrievers/verifiers might exploit.
	4.	The paper mentions integrating weights into reranking; however, it would help to give a more explicit formula and discuss alternatives (e.g., score calibration/fusion variants). The algorithm box gives the core update but not the full fusion equation.
	5.	The approach requires LLM verification per subclaim over top-k documents; reporting wall-clock and cost (per example) would inform practical deployment.

**Questions:**

1.	How sensitive is performance to stance classification accuracy? Could you report results using a weaker verifier and/or with controlled label noise injected into stance labels?
	2.	What happens when the ratio of misinformation is pushed beyond your default protocol (e.g., >70% of retrieved evidence for a subclaim)? Any catastrophic flips?
	3.	Please provide the exact scoring equation used for reranking (e.g., how normalized retrieval scores combine with the sigmoid-transformed weights).
	4.	Have you tried a non-Wikipedia KB (e.g., curated domain corpora) with less redundancy to test the reliance on redundancy assumptions?
	5.	Any qualitative analysis where the system wrongly down-weights true refutations (i.e., the claim is false, but the loop favors supporters)?

---

> ### Author Response · Authors · 2025-11-27
> **Respond to Reviewer Tfww**
>
> # To Reviewer Tfww
>
> We thank Reviewer Tfww for the comprehensive review and insightful advices.
>
> > The problem formulation implicitly assumes misinformation is a minority and truth is redundant (KB redundancy). While often reasonable for Wikipedia, the method’s behavior when misinfo ≥ truth (or redundancy is low) is less clear. Some ablations reduce contamination, but the extreme-noise regime and failure modes deserve more analysis.
>
> We already include cases with misinfo far more than gold evidence. In Table 3, misinfo is generally 2x - 9x more than gold evidence.
>
> > The reweighting hinges on LLM stance accuracy; systematic stance errors could amplify misinformation. A more explicit error-propagation analysis or confidence calibration for the verifier would strengthen the argument. (Figure/algorithm explain the mechanics but not robustness to verifier noise.)
>
> We agree that the innate ability of the LLM strongly affect our results. However, we do not directly use the prediction results from the LLM as stance prediction. Instead, the retrieval score and the weight score both contribute to the final stance prediction. Thus, even if a misinformation is wrongly upweighted at the beginning, it may be downweighted if it is later refuted by more redundant true documents.
>
> > The contamination strategy is topical and adversarial, but more discussion is needed on (a) how close it is to in-the-wild web noise, and (b) safeguards against inadvertently encoding dataset labels into style cues that retrievers/verifiers might exploit.
>
> We use two types of prompts to generate misinformation (see Appendix B2 and B3): (1) generating a contradictory claim to a given claim by specifying labeled entities and supporting evidence, and (2) generating “supportive” evidence for false claims.
> Strategy (1) is more direct and relatively rigid, while strategy (2) is more indirect and allows for more creativity. These two approaches are complementary and together increase the diversity of the generated misinformation, helping to better approximate the quality of in-the-wild web noise.
> In our experiments, we did not observe any systematic stylistic patterns that distinguish true from false information that could be inadvertently exploited by the retriever or verifier. Empirically, the retriever successfully retrieved over 95% of both gold evidence and misinformation. Moreover, the verifier does not show a tendency to judge truth or misinfo just by the document itself: the claim can either be true or false, and the verifier only perform stance grouping.
>
> > Reporting wall-clock and cost (per example) would inform practical deployment.
>
> We added wall-clock and cost (per example) to each dataset and each base LLM to the appendix A3 Table 7 of the updated rebuttal revision version.
>
>
> > How sensitive is performance to stance classification accuracy? Could you report results using a weaker verifier and/or with controlled label noise injected into stance labels?
>
> Our method performance strongly relies on the reasoning ability of the base LLM verifier, like previous approaches such as GraphCheck (Jeon & Lee, 2025) and ProgramFC (Pan et al., 2023). If the verifier failed to identity stances of documents towards the claim, the prediction will be misleading.
>
> > What happens when the ratio of misinformation is pushed beyond your default protocol? Any catastrophic flips?
>
> Actually, many wrong cases are due to overwhelming misinformation (See Table 3). Especially for PUBHEALTH and Climate-FEVER.
>
> > Please provide the exact scoring equation used for reranking.
>
> For each document $d$, we initialize two fields in the weight database $p$ for positive weight and $n$ negative weight, both with 0. Every time we perform a retrieval, we get a list of documents with their corresponding retrieval scores. Now, given in a retrieval, a document with current positive weight $p$, negative weight $n$ and original retrieval score $r$ (normalized to [0,1]), then the score used for reranking $r'$ is:
> $$
>     r' = r \cdot \frac{1}{1 + e^{n - p}}
> $$
>
> > Have you tried a non-Wikipedia KB with less redundancy to test the reliance on redundancy assumptions?
>
> Yes, in our experiments, PublicHealth, SciFact and Climate-FEVER are not queried on Wikipedia, and are much less redundant. The performances on these datasets have greater variances than the other three (See Table 1 and Table 4). Thus we believe that redundancy is important for misinfo fact checking.
>
> > Qualitative analysis where the system wrongly down-weights true refutations?
>
> False positive prediction usually happens when the fact is binary and the number of misinfo documents is more than that of true documents. In the 9th example in the hotpotQA, there is a FALSE claim: "Annie Morton is older than Terry Richardson". Our retrieval from misinfo knowledge base contains 4 misinfo documents saying Annie Morton is older than Terry Richardson, and only two true documents refuting. This results in a false positive prediction.

---

### Meta-Review · Area_Chair_gKpW · 2025-12-07

**Summary:**

The paper proposes a new approach for fact checking that is robust misinformation in the form of knowledge base contamination.  The reviewers' concerns include:
1. A lack of novelty since fact checking under contradictory information is not new.
2. Evaluation of the approach on artificially contaminated data that might simply align with the design of the proposed approach.
3. Assumption that there is less misinformation than correct information and simplistic technique that consists of a wighted majority vote
4. Heuristic nature of the proposed technique
5. No runtime and scalability evaluation

**Reviewer Concerns:**

The following concern was addressed:

5.  The paper was updated to report the running time

The following concern was partly addressed:

3. The authors clarified that the experiments included twice as much misinformation than correct information.  However, it remains unclear why the proposed technique worked since it is taking form of weighted majority vote.

**Reviewer Scores:**

Overall, I expect the reviewers' scores to remain the same because the rebuttal focused on providing explanations without necessarily addressing the most important concerns nor leading to changes in the paper that could provide additional evidence.

Reviewer Tfww:

I expect the reviewer to keep its score unchanged because the authors simply explained that the experiments do include twice as much misinformation as correct information, but it remains unclear why the approach works.  Since the approach is a form of weighted majority vote, one still expects that in situations with more misinformation than correct information that the approach would fail.  The authors explain that the misinformation lacks consistency and therefore will get down weighted, but it is not clear how this happens just based on the pseudocode in Algorithms 1, 2 and 3.  Overall, the approach consists of a heuristic without a clear understanding of its properties.

Reviewer 6kAj:

I expect the reviewer to keep its score since the authors failed to acknowledge previous work and to add a suitable related work section that discusses previous work about dealing with contradictory information and how the proposed work is different/novel.  The paper also failed to include an evaluation on natural data that includes misinformation (e.g., TREC benchmark mentioned by the reviewer).  Hence, there is a risk that the proposed approach works well simply because it was designed specifically to deal with the type of artificial contamination that was used for the evaluation.  Finally, similar to the previous reviewer, the approach is a heuristic and it is not clear why it works when there is more misinformation than correct information.

Reviewer UwZ8:

I expect the reviewer to keep its positive score since the authors addressed the concern of scalability by adding a new table that reports the running time.  The questions about claim processing order and quantity of misinformation were answered adequately.

Reviewer oMqg:

I expect the reviewer to keep its score since its main concerns were not addressed.  The problem statement and proposed approach remain ad hoc/heuristics and it is not clear why the proposed approach should work when there is more misinformation than correct information.

---

### Decision · Program_Chairs · 2026-01-26

Reject